# Exploring the Impact of the Turning of AISI 4340 Steel on Tool Wear, Surface Roughness, Sound Intensity, and Power Consumption under Dry, MQL, and Nano-MQL Conditions

Yusuf Fedai

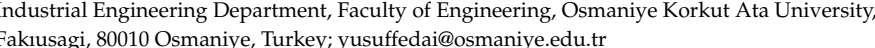

Industrial Engineering Department, Faculty of Engineering, Osmaniye Korkut Ata University, Fakıusagi, 80010 Osmaniye, Turkey; yusuffedai@osmaniye.edu.tr

**Abstract:** Optimizing input parameters not only improves production efficiency and processing quality but also plays a crucial role in the development of green manufacturing engineering practices. The aim of the present study is to conduct a comparative evaluation of the cutting performance and machinability process during the turning of AISI 4340 steel under different cooling conditions. The study analyzes cutting operations during turning using dry, minimum quantity lubrication, and nano- minimum quantity lubrication. As control parameters in the experiments, three different cooling types, cutting speeds (100, 150, 200 m/min), and feed rate (0.1, 0.15, 0.20 mm/rev) levels were applied. The experimental results show that the optimal output values are found to be Vb = 0.15 mm, Ra = 0.81 μm, 88.1 dB for sound intensity and I = 4.18 A for current. Moreover, variance analysis was performed to determine the effects of input parameters on response values. Under dry, minimum quantity lubrication, and nano-minimum quantity lubrication processing conditions, parameters affecting tool wear, surface roughness, current by the motor shaft, and sound level were examined in detail, along with the chip morphology. The responses obtained were optimized according to the Taguchi S/N method. As a result of optimization, it was concluded that the optimum values for cutting conditions were nano-minimum quantity lubrication cooling and V = 100 m/min, f = 0.1 mm/rev cutting. Finally, it was observed that there was a 13% improvement in tool wear, 7% in current, 9% in surface roughness, and 8% in sound intensity compared to the standard conditions. In conclusion, it was determined that nano-minimum quantity lubrication with the lowest level of cutting and feed rate values provided the optimum results.

**Keywords:** CNC turning; AISI 4340; nano-MQL; current; sound intensity; chip morphology





## 1. Introduction

AISI 4340 steel, which contains elements such as nickel, molybdenum, and chromium in different proportions, is a commercial steel known for its good corrosion resistance, hardenability, ductility, and high strength [1]. It finds applications in various fields such as the aerospace industry, automotive industry, power plants, maritime and defense industries, petroleum and gas industry, and weapons industry, especially after undergoing different heat treatments [2]. Considering the areas of use of this material, it is classified as a challenging alloy due to its high strength and low thermal conductivity. Consequently, excessive heating occurs in the machining zone, leading to issues such as poor surface finish and excessive tool wear. Cooling of the machining zone is of paramount importance to address the problems caused by excessive heat. Many studies have been conducted to optimize cutting parameters of AISI 4340 alloy steel and improve the quality of the machined surface [3–6]. In these studies, researchers have focused on reducing surface roughness to ensure the integrity of the machined surface, minimizing tool wear and the resulting increase in cutting forces, reducing power consumption, enhancing machining efficiency, optimizing cutting parameters such as cutting speed and feed rate, as well as controlling pa-

rameters such as cooling and lubrication methods, and exploring environmentally friendly green manufacturing [7].

The limited availability of resources worldwide necessitates optimization in production and environmental aspects. However, the increasing demand for prosperity in developing societies leads to increased resource consumption. As resource consumption rises, it becomes imperative to minimize environmental pollution and harmful emissions. Similarly, in the field of metalworking, particularly in chip manufacturing, the growing importance of human health and environmental awareness necessitates new studies on the use of cutting fluids [8]. Compared to traditional methods, the cost of lubricating/cooling cutting fluids is estimated to be around 17% of the total production costs within machining. Besides the high cost, the environmental and health hazards associated with these fluids have prompted researchers to explore new cooling techniques [9,10].

Traditionally, abundant cutting fluid is used when machining certain materials. However, this approach increases component costs and energy consumption. Therefore, researchers have found this a compelling reason to explore alternative lubrication–cooling strategies to enhance cutting performance. In a study by Gong et al. [11], machining of Inconel 718 alloy was carried out under cryo, dry, wet, MQL, and nanofluid MQL conditions. Minimum quantity lubrication (MQL) refers to the mixture of pressurized air in the form of fine droplets that atomize and create a spray of a small quantity of lubricating liquid in the cutting zone. The intensity and quantity of the aerosol cloud can be controlled using various valves [12]. The application of MQL in machining operations has demonstrated various benefits, including limited environmental impact compared to the abundant use of traditional cutting fluids, reduced production costs, and increased worker safety. However, MQL has its limitations in terms of cooling function due to the inability to completely limit heat generation in both primary and secondary machining zones, mainly because of the lower oil flow rate. Hence, there is a need to enhance cutting performance in MQL processes, and hybrid nanofluid-assisted MQL applications have recently become important research trends to improve MQL efficiency [13,14]. For a lubrication system using MQL and nano-MQL to be effective in cutting, the quantity of liquid delivered to the work zone per minute and the air pressure, as well as the type and quantity of nano material added to the cutting fluid, are important parameters. Patole and Kulkarni optimized cutting parameters by adding 0.02% MWCNT to the cutting fluid during the turning of AISI 4340 steel. The results indicated that the nano-fluid coolant was highly dependent on the parameters, with flow rate and pressure following suit. The optimal values for these parameters were determined to be 5 bars of pressure and a flow rate of 140 mL/min. They claimed that using MWCNT-doped nano-MQL systems could reduce tool wear and achieve good surface roughness [15]. Palanisamy et al. investigated the effect of applying high-pressure cutting fluid to the cutting zone during the machining of titanium alloys. They achieved better surface quality on the machined material and a longer tool life. The study also revealed that cooling fluids at different pressures caused significant changes in chip morphology [16]. In a study by Ramanan et al., nano fluids with higher thermal conductivity compared to traditional cutting fluids were used in the MQL system to investigate their effect on cutting parameters [17].

Studies on cooling fluids with minimal environmental impact and workplace safety concerns continue from the perspective of operators [18]. In this regard, Chetan et al. compared $Al_2O_3$ nano particle-based nMQL with cryogenic cooling during the turning of Nimonic 90 alloy [19].

Modern machining industries primarily prioritize factors such as workpiece dimensional accuracy, surface finish, cutting temperature, high production rates, extended cutting tool lifespan, cost savings, occupational health, machining performance, and energy consumption reduction, especially with a focus on environmental concerns. Surface roughness and noise pollution are crucial factors, and various studies in the literature have demonstrated that machining under minimum quantity lubrication (MQL) and nano-MQL condi-

tions leads to improvements in energy consumption, surface roughness, sound intensity and cutting forces [20–22].

Recently, numerous researchers have investigated the balance between cutting quality and power consumption during machining operations [23]. The continuous rise in energy demand and the constraints associated with increasing carbon emissions have exerted significant pressure on manufacturing industries to save energy. In one study, Abbas et al. explored the surface roughness and power consumption of AISI 1045 steel when machined in a nanofluidized MQL environment [24].

Sound levels during machining processes are parameters of particular concern for occupational health and safety. In the machinery manufacturing industry, especially in large facilities with numerous machine tools in close proximity, machine operators are exposed to high levels of noise. It is essential to mitigate noise levels in this context. Svenningsson and Tatar investigated sound generation using different cutting methods and inserts in a study on AISI 4340 material. Through their analysis and measurements, they proposed that the source of the sound is related to the vibration mode of the chip. Chip segmentation influences cutting forces, thereby increasing the current [25]. Albayrak et al. investigated the effects of cutting parameters, including feed rate, spindle speed, and chip depth, at three levels, on the sound level and surface roughness during the turning of SAE 4140 alloy steel [26]. In a study by Downey et al., the machining precision, tool wear performance, and surface roughness of AISI 4340 material were experimentally compared using the Acoustic Signals (AS) method with a high-speed steel (HSS) cutting tool and a physical vapor deposition (PVD) titanium carbon nitride (TiCN) insert. It was reported that as tool wear progressed over time, the emitted sound changed, and a unique sound signature characterizing wear was observed for each wear phase [27].

As a result of the literature studies conducted, it has been observed that there is a notable lack of research on various aspects of AISI4340 alloy steel machining, including the cooling/lubrication conditions during machining, environmental consciousness in conjunction with green production, the impact of noise generated during work on both human health and cutting parameters, wear, surface roughness, and energy consumption. This study aims to contribute to previous research by statistically evaluating the reciprocal effects of dry, MQL, and nano-MQL cooling under different machining conditions, focusing on chip morphology. The study employs the Taguchi signal-to-noise (S/N) method to optimize tool wear, power consumption, sound level reduction, and surface roughness.

## 2. Materials and Methods

### 2.1. CNC Lathe, Workpiece and Cutting Tool

For the turning experiments of AISI 4340 steel, workability criteria such as surface roughness, tool wear, sound intensity, and the current during machining were measured. The experiments were conducted using a Yunnan Cy-K360n CNC Lathe with a maximum of 10,000 rpm and a 7.5 kW main motor power. Commercial AISI 4340 (DIN 34CrNiMo6) steel, in its as-received condition without any physical or chemical treatments (e.g., heat treatment, etc.), was used as the workpiece material. The round bar-shaped AISI 4340 (DIN 34CrNiMo6) steel was cut using a band saw with a cooling fluid to obtain dimensions of 360 mm in length and Ø135 mm in diameter. The chemical properties of the AISI 4340 alloy steel used in the experiments are provided in Table 1, while the mechanical properties are given in Table 2 [28]. The cut workpiece was prepared for the experiments by performing surface and face turning to remove surface impurities and oxide layers, ensuring that the workpiece was clean before commencing the experiments. All experiments involved the machining of a Ø135 diameter. The CNC lathe machine and the workpiece used in the experiments are depicted in Figure 1.

**Table 1.** Chemical composition of AISI 4340 (34CrNiMo6) steel.

| Element | C | Mn | P | S | Si | Ni | Cr | Mo | Fe |
|---|---|---|---|---|---|---|---|---|---|
| %Wt. | 0.3–0.38 | 0.50–0.80 | ≤0.025 | ≤0.035 | ≤0.40 | 1.3–1.7 | 1.3–1.7 | 0.15–0.30 | Rest |

**Table 2.** Mechanical properties.

| Property | Symbol | Value |
| --- | --- | --- |
| Ultimate Tensile Strength | $\sigma_{UTS}$ (Mpa) | 1210 |
| Yield Strength | $\sigma_{Ys}$ (Mpa) | 1084 |
| Rupture Strain | A (%) | 12.2 |
| Reduction in The Area | Z (%) | 60.2 |
| Young's Modulus | E (Gpa) | 210 |

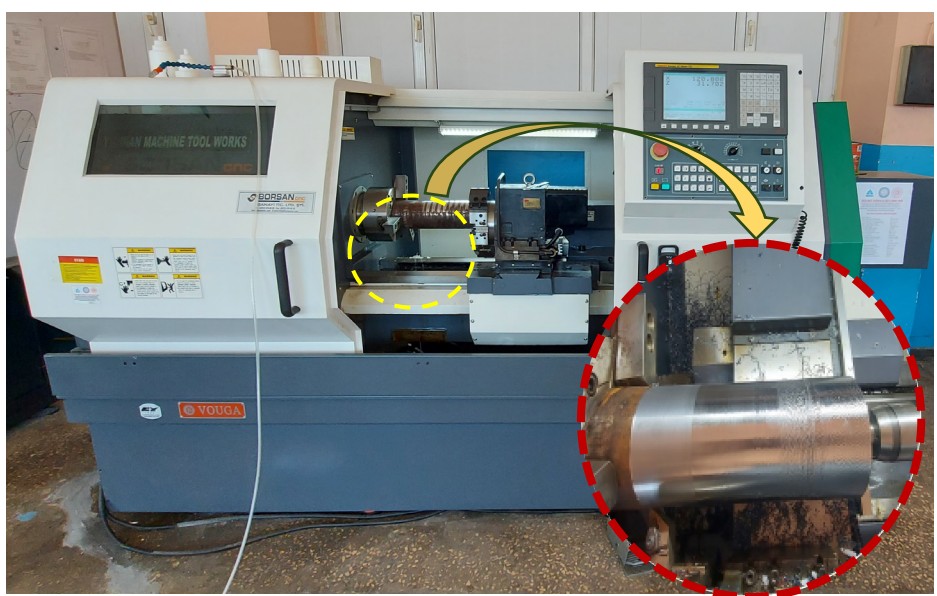

**Figure 1.** CNC lathe and work piece.

In the experiments, PVD-coated cutting inserts of TaeguTec brand, DNMG 150608 TT5080 type, were used. For each experiment, one face of the cutting insert was utilized and recorded for wear measurement. The cutting insert has approximate dimensions of 15 mm in width, 6 mm in thickness, and a corner radius of 0.8 mm. The dimensions and appearance of the cutting insert are provided in Figure 2.

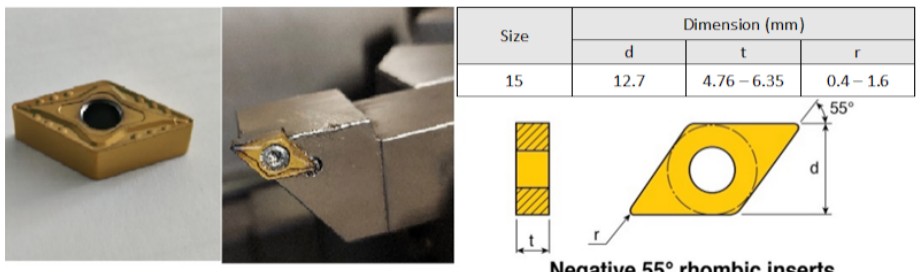

**Figure 2.** Cutting insert dimensions.

### 2.2. MQL System and Preparation of Nano Fluid

The primary objective of the MQL (minimum quantity lubrication) system is to achieve optimal chip removal from the workpiece under environmentally friendly conditions, taking into consideration factors such as cost, health, and the environment. In this process, water-based biodegradable lubricants, which are environmentally friendly, are atomized and sprayed into the cutting area under high pressure, reducing the heat generated between the chip and the tool and carrying it away from the cutting zone. In the experiments, the S.B.H. STN 15 MQL system, which operates within a range of 4–6 bar air pressure as shown in the schematic representation in Figure 3, was used. The MQL flow rate varies in the

literature between 60 and 300 mL/h. Several studies have reported improved surface roughness at higher flow rates [29]. In accordance with the literature [30], a flow rate of 100 mL/h was used in the experimental study.

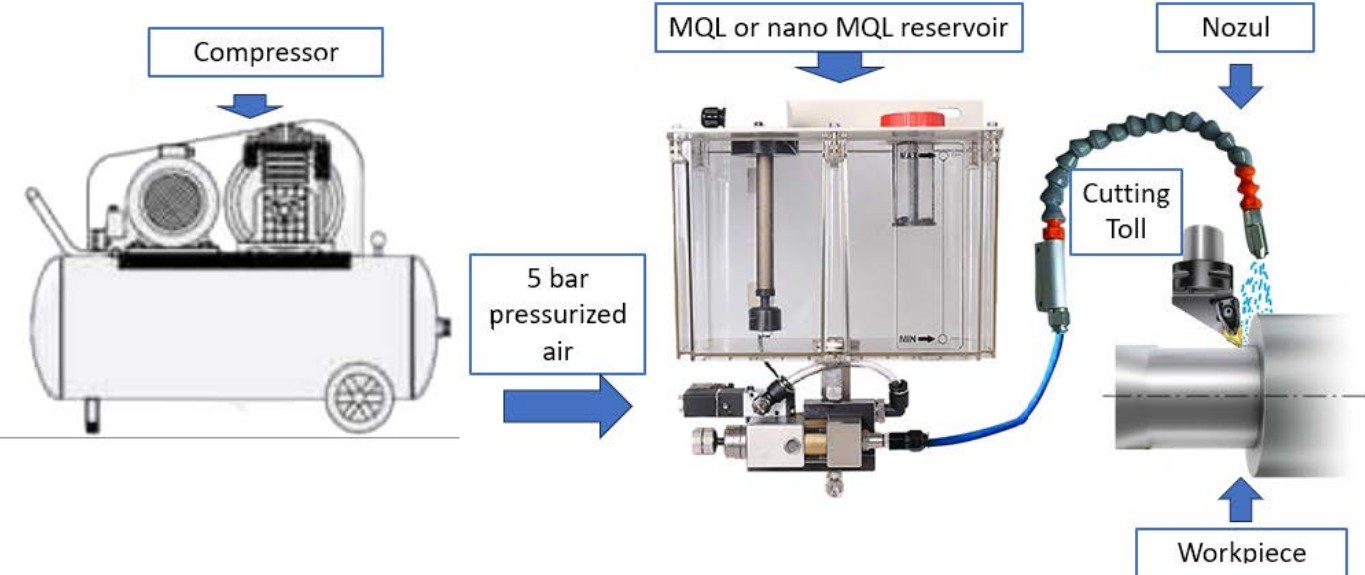

**Figure 3.** Schematic MQL system.

### 2.3. Preparation of MQL System and Nano Fluid

Nano-fluids are a new class of fluids obtained by mixing nano-sized materials into the coolant fluid. In this study, multi-walled carbon nanotubes (MWCNT) with a diameter of 7 nm and a length of 5 μm were mixed with the cutting fluid. Triethanolamine was used as the cutting fluid. Triethanolamine is an environmentally friendly choice as it is a water-based chemical compound. Mixing was carried out by adding 1% by weight of MWCNT to 1 L of cutting fluid. The mixing process of the nanomaterial into the cutting fluid occurred in three stages (Figure 4). In the first stage, the nanomaterial was weighed and then introduced into the cooling fluid, followed by mechanical stirring at 750 rpm for 1 h. This step was aimed at preventing the clumping of the nanomaterial within the liquid. In the second stage, ultrasonic mixing was performed for 1 h, and in the third stage, magnetic stirring was carried out at 1500 rpm for 2 h to prevent the settling of the nanomaterial. The mixed nanomaterial was directly used in the experiments to prevent any settling.

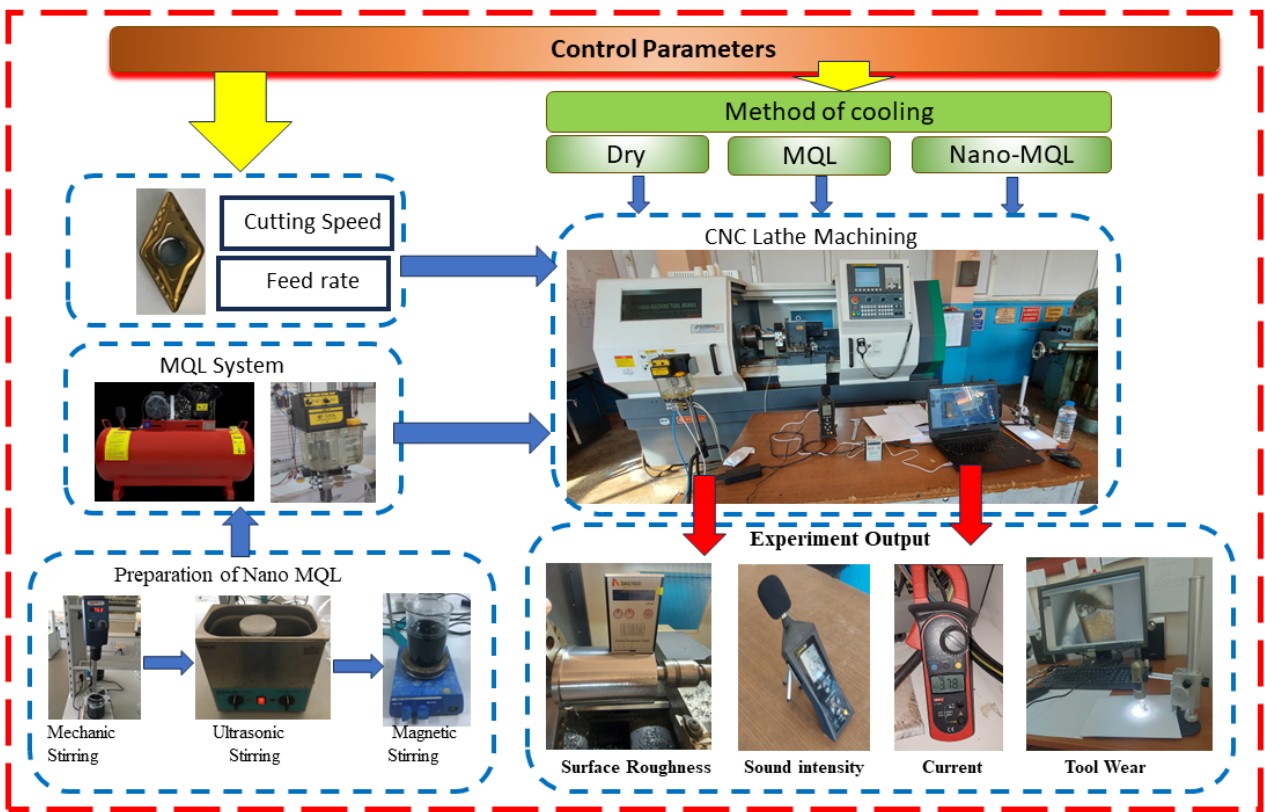

**Figure 4.** Experimental setup of the study.

### 2.4. Measurement of Experimental Outputs

Surface roughness is one of the essential factors in machining operations on metals. The quality of the machined surface is significantly represented by surface roughness. In this study, 'Ra' was considered as the roughness parameter. Surface roughness values were measured using the Dailyaid DR100 model roughness measurement device. For these values, the arithmetic average of measurements taken from three different points immediately after the machining experiment was calculated. The samples were thoroughly cleaned with air before measurements. The measurement device was axially placed on the workpiece. To reduce measurement errors, three different points were selected along the same axis for all experiments. Sampling length and evaluation length for surface roughness values were set at 0.8 mm and 4 mm, respectively.

Sound is a significant factor in terms of environmental pollution. Therefore, in the experiments, the sound generated by the machine and during cutting was selected as a parameter. To obtain reliable data during measurements, ambient sounds were isolated. Initially, the CNC lathe was run without any cutting to determine the background noise level, and values above this level were identified as cutting sound data. PCE brand and 322A model sound measurement devices were used for measurements. The device has a measurement capacity within the frequency range of 30–130 dB and can capture data in the range of 1 s to 125 milliseconds.

The current value was measured using a UNI-T brand and UT202 model clamp ammeter. The device allows high-precision measurements with an extended current frequency. It has a 4000-count display and data-holding function, facilitating the analysis of measurement data. Due to its design, the clamp ammeter can only measure a single-phase current. Therefore, the phase going to the main spindle of the lathe was identified, and the clamp ammeter was mounted on this cable. During the experiment, the average of the varying current values was calculated.

The measurement of the cutting tool wear amounts was performed using a Dino-Lite digital microscope. After a specific length of cutting operation, the cutting tool was

removed, and photographs of the worn surfaces of the cutting tools were taken with the microscope to measure the side wear amounts.

Detailed images of the control parameters and outputs for the CNC used in the experiments are provided in Figure 4.

*2.5. Cutting Condition and Design of Experiment*

During chip removal operations, a significant amount of electrical energy is primarily consumed, depending on various cutting parameters. In this study, the impact of input parameters was investigated with a focus on environmental consciousness and minimizing power consumption during machining. Particularly, as outputs of an environmentally friendly machining operation, surface roughness, tool wear, and, current, sound intensity—a crucial factor for the environment—were selected as output parameters in the optimization of control parameters. The control parameters selected included cooling of method (MOC) cutting speed and feed rate, taking into account the influence of environmental factors on methods for cooling the cutting zone. The levels of the fundamental cutting parameters were chosen in accordance with the recommendations of the tool manufacturer and existing literature. Preliminary experiments were conducted to test the interaction between the machine, cutting tool, and workpiece. The selected parameters and levels are shown in Table 3.

**Table 3.** Process parameters and their levels.

| Control Parameters | Notation | Levels of Factors | | |
|---|---|---|---|---|
| | | Level 1 | Level 2 | Level 3 |
| Method of Cooling (-) | MOC | Dry | MQL | Nano-MQL |
| Cutting Speed-(m/min) | V | 100 | 150 | 200 |
| Feed Rate-(mm/rev) | f | 0.1 | 0.15 | 0.2 |

Experimental design is a method used to plan and conduct experiments. The Taguchi orthogonal array is a widely used statistical method for the analysis of process and product improvements. With this method, the best factors for obtaining the most optimal results with a small number of experiments can be determined. The Taguchi method has significant potential for the cost-effective analysis of experiments. Therefore, experiments were conducted using Taguchi's L9 orthogonal array(Table 4), which uses three factors at three levels, in order to achieve the best results from a series of experiments [31,32]. The impact levels of variables on the outputs were determined by applying variance analysis (ANOVA) to the experimental results with a 95% confidence interval. The experimental design and statistical analyses, according to the Taguchi method, were carried out using Minitab 20 software.

**Table 4.** Taguchi L9($3^3$) orthogonal array.

| Exp. No. | MOC | V (m/min) | f (mm/rev) | MOC | V | f |
|---|---|---|---|---|---|---|
| 1 | Dry | 100 | 0.1 | 1 | 1 | 1 |
| 2 | Dry | 150 | 0.15 | 1 | 2 | 2 |
| 3 | Dry | 200 | 0.2 | 1 | 3 | 3 |
| 4 | MQL | 100 | 0.15 | 2 | 1 | 2 |
| 5 | MQL | 150 | 0.2 | 2 | 2 | 3 |
| 6 | MQL | 200 | 0.1 | 2 | 3 | 1 |
| 7 | Nano-MQL | 100 | 0.2 | 3 | 1 | 3 |
| 8 | Nano-MQL | 150 | 0.1 | 3 | 2 | 1 |
| 9 | Nano-MQL | 200 | 0.15 | 3 | 3 | 2 |

In this method, a statistical performance measure known as the signal-to-noise (S/N) ratio is used to analyze the results. The results obtained from the experiments are converted into signal-to-noise (S/N) ratios for evaluation. In the calculation of S/N ratios,

three different methods, known as larger-the-better, smaller-the-better, and nominal-the-best, are used depending on the characteristic type. In determining the S/N values in this study, it was desired to minimize the SI values for sound intensity, minimize the (Ra) surface roughness for machining efficiency, minimize tool wear (Vb), and minimize power consumption represented by I (A). Therefore, the formula corresponding to the "smaller-the-better" principle given in Equation (1) was used [33].

$$\text{smaller is better;} \quad \frac{S}{N} = -10\log\left[\frac{1}{n}\sum_{i=1}^{n}y_i^2\right] \tag{1}$$

The objective here is to minimize the noise function, in other words, maximize the S/N ratio. Therefore, in the evaluations, the level with the highest S/N ratio among the calculated average S/N ratios for each parameter is used to determine the best result.

## 3. Result and Discussion

### 3.1. Effects of Process Parameters on Outputs

In this section, the effects of cutting speed and feed rate on the outputs in the turning of AISI 4340 steel under cooling conditions of dry, MQL, and MQL with MWCNT additive have been examined. For all experimental trials after turning operations, the measured tool wear, surface roughness, sound intensity, and the power consumption values calculated for the machining process, corresponding to the processed surfaces, are shown in Table 5. The interaction status of input parameters on the outputs obtained from the experiments has been compared using three-dimensional graphics. In the study, experimental results were analyzed using ANOVA to determine the impact levels of control factors on the outputs. Response variables were transformed into signal-to-noise (SN) ratios in the Taguchi method. The calculated Taguchi Signal/Noise ratios are also given in Table 5. As a result of the experiments, the values minimized as expected were observed in the experiments conducted with nano-MQL cooling, except for power consumption. All experiments, were minimized under condition $A_3B_2C_1$, which corresponds to experiment number 8. These values occurred as Vb = 0.15 mm, Ra = 0.81 μm, 88.1 dB for sound intensity, and I = 4.18 A for the current. This situation is also evident from the fact that the max. values of the S/N ratio are the same as the best values.

**Table 5.** Experimental result and S/N ratio.

| Exp. No. | MOC | V (m/min) | f (mm/rev) | Vb (mm) | S/N Vb | Ra (μm) | S/N Ra | SI (dB) | S/N SI | I (A) | S/N I |
|---|---|---|---|---|---|---|---|---|---|---|---|
| 1 | 1 | 100 | 0.1 | 0.241 | 12.36 | 1.35 | −2.607 | 94.3 | −39.5 | 5.13 | −14.2 |
| 2 | 1 | 150 | 0.15 | 0.289 | 10.78 | 1.48 | −3.405 | 98.7 | −39.9 | 6.65 | −16.4 |
| 3 | 1 | 200 | 0.2 | 0.42 | 7.53 | 2.12 | −6.527 | 101.3 | −40.1 | 7.21 | −17.2 |
| 4 | 2 | 100 | 0.15 | 0.218 | 13.23 | 1.22 | −1.727 | 95.2 | −39.6 | 5.45 | −14.7 |
| 5 | 2 | 150 | 0.2 | 0.27 | 11.37 | 1.91 | −5.621 | 97.6 | −39.8 | 5.64 | −15.0 |
| 6 | 2 | 200 | 0.1 | 0.205 | 13.76 | 1.1 | −0.828 | 93.7 | −39.4 | 5.01 | −14 |
| 7 | 3 | 100 | 0.2 | 0.217 | 13.27 | 1.36 | −2.671 | 92.3 | −39.3 | 5.11 | −14.1 |
| 8 | 3 | 150 | 0.1 | 0.15 | 16.48 | 0.81 | 1.83 | 88.1 | −38.9 | 4.18 | −12.4 |
| 9 | 3 | 200 | 0.15 | 0.169 | 15.44 | 1.05 | −0.424 | 90.1 | −39.1 | 4.52 | −13.1 |
| | | | Min | 0.15 | 7.53 | 0.81 | −6.527 | 88.1 | −40.11 | 4.18 | −17.2 |
| | | | Max | 0.42 | 16.48 | 2.12 | 1.83 | 101.3 | −38.9 | 7.21 | −12.4 |

### 3.2. Analysis of Variance (ANOVA)

In statistical analysis, especially in the field of engineering, analysis of variance (ANOVA) is commonly used to evaluate experimental data. The aim of ANOVA is to determine to what extent the factors under investigation influence the selected output values that measure quality [34]. In this research, the direct interaction of control factors (MOC, V, f) on the outputs was analyzed using ANOVA. In the ANOVA table, when the

*p*-value is less than 0.05, the regression equation and the examined factors are considered statistically significant. In addition, the percentage contribution ratio (PCR) of the terms in the estimated model to the total variation can be checked to assess the degree of influence of the factors on the model. The evaluations performed by PCR are described in detail in Figure 5. The results of the variance analysis calculated for all outputs are given in Table 6. The table shows the F values and the percentage contribution ratio (PCR) indicating the significance level of each variable. This analysis was performed with 95% confidence interval and 5% significance levels. The effect of control factors is determined by comparing the F values. The factor with the highest F value has the greatest impact on the result. The significance of the results is determined by the *p*-value. A *p*-value less than 0.05 indicates that the factor is statistically significant. It can be seen from the table that for tool wear, MOC and f values are both less than 0.05. The *p*-values indicating statistical significance are $0.002 \leq 0.05$ for MOC and $0.008 \leq 0.05$ for feed rate, making them statistically significant. Cutting speed, on the other hand, is greater than 0.05, indicating it is not significant.

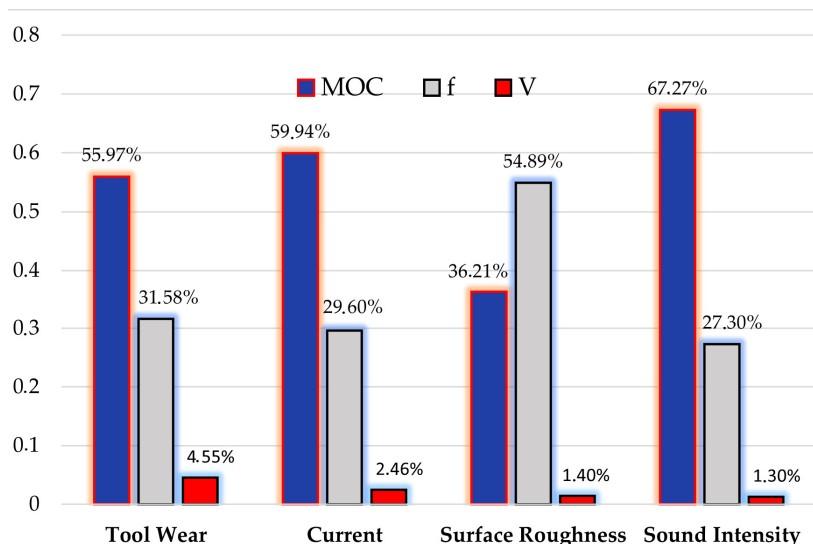

**Figure 5.** ANOVA percentage contribution ratio (%PCR).

**Table 6.** Variance analysis (ANOVA) for turning responses.

| | Source | DF | PCR% | Adj SS | Adj MS | F-Value | *p*-Value | Significant |
|---|---|---|---|---|---|---|---|---|
| | MOC | 2 | 55.97 | 0.029 | 0.029 | 35.41 | 0.002 | Yes |
| | V | 2 | 4.55 | 0.002 | 0.002 | 2.88 | 0.151 | No |
| Tool Wear | f | 2 | 31.58 | 0.016 | 0.016 | 19.98 | 0.007 | Yes |
| (Vb) | Error | 18 | 7.90 | 0.00403 | 0.00081 | | | |
| | Total | 26 | 100.00 | | | | | |
| | S = 0.0284 | | $R^2$ = 92.10% | | | $R^2$(adj) = 87.35% | | |
| | MOC | 2 | 59.94 | 4.472 | 4.472 | 37.48 | 0.002 | Yes |
| | V | 2 | 2.46 | 0.184 | 0.184 | 1.54 | 0.270 | No |
| Current | f | 2 | 29.60 | 2.208 | 2.208 | 18.51 | 0.008 | Yes |
| (I) | Error | 18 | 8.00 | 0.597 | 0.119 | | | |
| | Total | 26 | 100 | | | | | |
| | S = 0.03454 | | $R^2$ = 92.00% | | | $R^2$ (adj) = 87.21% | | |
| | MOC | 2 | 36.21 | 0.499 | 0.499 | 24.14 | 0.004 | Yes |
| | V | 2 | 1.40 | 0.019 | 0.019 | 0.93 | 0.379 | No |
| Surface Roughness | f | 2 | 54.89 | 0.75615 | 0.75615 | 36.59 | 0.002 | Yes |
| (Ra) | Error | 18 | 7.50 | 0.10332 | 0.02066 | | | |
| | Total | 26 | 100 | | | | | |
| | S = 0.1437 | | $R^2$ = 92.5% | | | $R^2$ (adj) = 88.00% | | |

**Table 6.** *Cont.*

| | Source | DF | PCR% | Adj SS | Adj MS | F-Value | *p*-Value | Significant |
|---|---|---|---|---|---|---|---|---|
| | MOC | 2 | 67.27 | 94.799 | 94.799 | 81.53 | 0.000 | Yes |
| | V | 2 | 1.30 | 1.836 | 1.836 | 1.58 | 0.264 | No |
| Sound Intensity | f | 2 | 27.30 | 38.473 | 38.473 | 33.09 | 0.002 | Yes |
| (SI) | Error | 18 | 4.13 | 5.814 | 1.163 | | | |
| | Total | 26 | 100 | | | | | |
| | S = 1.078 | | $R^2$ = 95.87% | | | $R^2$ (adj) = 93.40% | | |

When examining the ANOVA table for surface roughness, the *p*-values calculated for MOC and feed rate are less than 0.05, indicating that these factors are statistically and physically significant in terms of surface roughness. The *p*-value for cutting speed is greater than 0.05, indicating that this parameter has no significant effect on surface roughness. This situation is clearly seen in the F and *p* values. For MOC, $p = 0.002 \leq 0.05$ and $F = 0.007 \leq 0.05$ are significant, while $V = 0.151 \geq 0.05$ is insignificant.

Percentage contribution ratio displays the percentage that each source in the analysis of variance table contributes to the total sequential sums of squares. As seen in Figure 5, the factor with the highest contribution ratio for tool wear is MOC, accounting for 55.97%, followed by feed rate with 31.58%, and cutting speed with 4.55%. With the current, MOC is the most effective parameter, with a contribution ratio of 59.94%, followed by feed rate with 29.6%, and cutting speed with 2.46%. According to the graph, the factor with the highest contribution ratio for surface roughness is the feed rate at 54.89%, followed by MOC with 36.21%, and cutting speed with 1.40%. This situation clearly demonstrates the effect of feed rate on surface roughness, in line with the literature. Finally, when looking at the PCR ratios for sound intensity, it is again evident that MOC is the most effective parameter, with 67.27%. It is followed by feed rate with 27.30%, while cutting speed has the least effect among all outputs, with a ratio of 1.30%. Overall, it can be observed that MOC is highly influential on all outputs, followed by feed rate with a significant impact, while cutting speed has almost no effect.

When conducting regression analysis, the S value is an important tool for assessing the quality of the model. A lower S value may indicate that the regression model explains the data better and that the predictions are more reliable. S represents the standard deviation of the distance between the data values and the fitted values. S is used to assess how well the model describes the response. The lower the value of S, the better the model describes the response. From Table 6, it can be observed that the lowest S value belongs to the team wear model, followed by current and surface roughness, respectively. It is also noted that the model with the highest S value is sound density.

The regression model is typically used to predict responses, representing the mathematical expression of the regression line obtained from the responses. It is used in conjunction with the error term to establish the relationship between responses and predictive parameters. First-degree regression equations modelling how the output parameters, namely tool wear, current, surface roughness, and sound intensity, change depending on input parameters have been provided in Equations (2)–(5), considering only main factor effects, based on the statistical and Taguchi analyses conducted.

$$Vb = 0.1656 + 0.0690\text{MOC} + 0.000393V + 1.037f \tag{2}$$

$$I = 4.815 - 0.863\text{MOC} + 0.0035V + 12.13f \tag{3}$$

$$Ra = 0.719 - 0.2883\text{MOC} + 0.00113V + 7.1f \tag{4}$$

$$SI = 93.28 - 3.975\text{MOC} + 0.01106V + 50.64f \tag{5}$$

$R^2$ is a statistical measure that gauges the success of a regression analysis. The $R^2$ value indicates how much of the variance in the dependent variable is explained by

the independent variables. $R^2$ takes a value between 0 and 1, where 0 means that the independent variables do not explain the dependent variable at all, and 1 means that the independent variables completely explain the dependent variable. In other words, a high $R^2$ value indicates that the regression model fits the data well, while a low $R^2$ value indicates a weak model. Adjusted $R^2$ is similar to $R^2$ but is used in regression models with multiple independent variables. Adjusted $R^2$ also takes a value between 0 and 1 [35].

The multiple regression coefficients of these first-degree equations have been modeled with high accuracy, at a 95% significance level, as follows: $R^2$ = 92.10%, $R^2$ (adj) = 87.35 for tool wear, $R^2$ = 92.00%, $R^2$ (adj) = 87.21 for current, $R^2$ = 92.50%, $R^2$ (adj) = 88.00 for surface roughness, and $R^2$ = 95.87%, $R^2$ (adj) = 93.40 for sound intensity. These values being greater than 80% indicate that the regression model fits well, and models close to reality can be obtained [36,37].

*3.3. Tool Wear*

Tool wear is the phenomenon where a tool experiences material loss and deformation in its structure due to the applied load on the cutting edge. Cutting parameters such as feed rate, cutting speed, and the cooling status of the cutting environment play a role in determining the level of tool wear, in addition to the characteristics of the workpiece and cutting tool. Flank wear is commonly used to determine the level of wear and is the most effective method used to predict tool life. The wear levels of the tools used in the experiments were measured, and their appearances were imaged under a microscope. When we examine the pictures according to the amount of wear, we observe that the highest wear occurred under the conditions of V = 200 m/min and f = 0.2 mm/rev with Vb = 0.420 mm (Red line), which corresponds to experiment number 3. The lowest wear, on the other hand, was observed under the conditions of V = 150 m/min and f = 0.1 mm/rev with Vb = 0.150 mm (Yellow Line), which occurred in experiment number 8 under nano-MQL conditions. When examining the images, it can be observed that non-uniform wear types such as chipping and breakage were not present in all tools, as shown in Figure 6. Burn marks were observed on the edges of the tools in dry cutting conditions due to the high heat generated. It is evident that this burn mark is significantly reduced in experiments conducted with MQL and nano-MQL. This can be attributed to the better cooling and lubrication properties provided by the nano cutting fluid. The limitation of wear only to flank wear on the tool edges can be attributed to the better conductivity, convection, and wetting properties of the nano cutting fluid [2].

The interaction of the input parameters obtained from the experiments on tool wear is presented in Figure 7. According to the figure, when each variable is analyzed, it can be seen that feed and cutting speed have the highest effect on tool wear, especially in dry-cutting conditions. According to Figure 7a,b, the tool wear value Vb shows a slight increase, in parallel with the values of f (0.10–0.15) and V (100–150) according to the change in MOC from 3 to 1. Under MOC 1 (dry cutting) conditions, the curves of both f and V have a similar slope, the amount of wear increases as the cutting and feed increases, and the greatest amount of wear occurs at the highest levels of both. This can be explained by the high heat generated between the tool and the workpiece under dry-cutting conditions, which significantly increases tool wear. In addition, for all parameters containing nano-MQL, less wear was observed at low speeds, but a slight increase in the wear value was observed as the speed and feed increased [38]. This finding is consistent with other studies conducted on the subject [39]. From Figure 7b,c, it can be observed that the lowest tool wear occurs at the medium cutting speed level of 150 m/min.

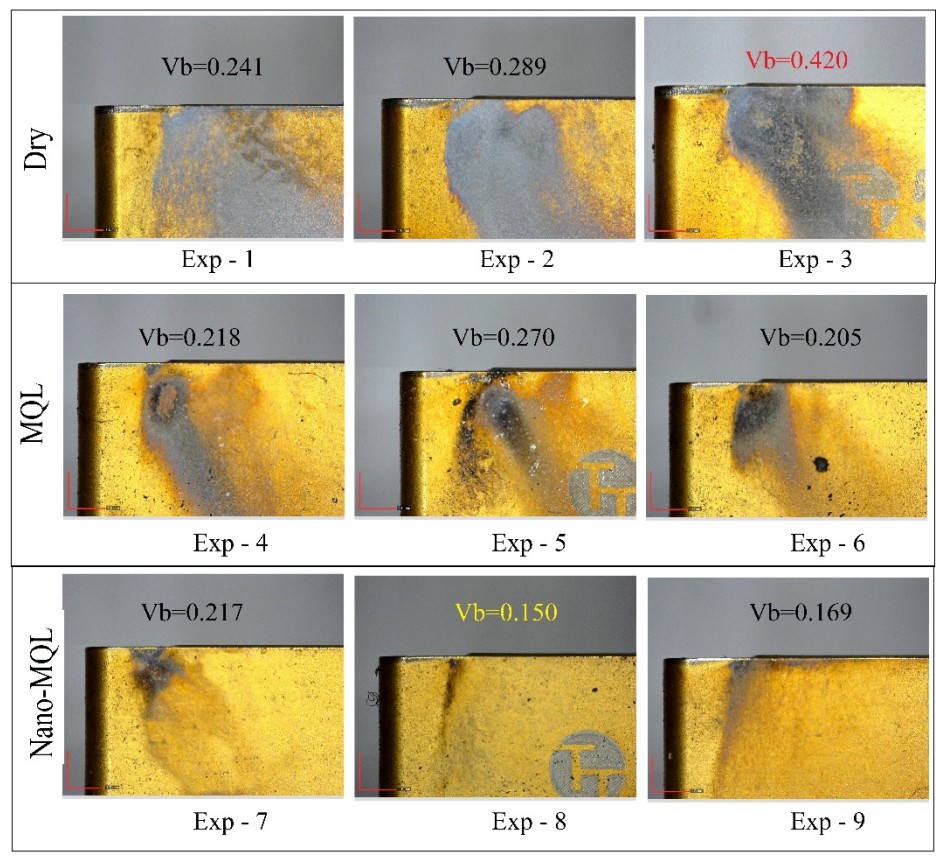

**Figure 6.** Microscopic images of tool wear produced during the turning by the dry, MQL, and nano-MQL lubrication techniques.

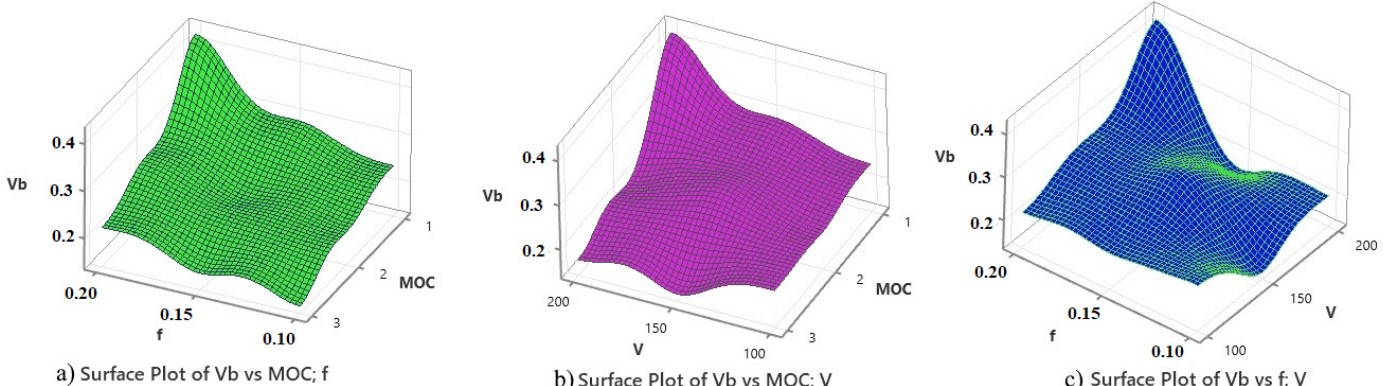

**Figure 7.** Three-dimensional graphs illustrating the effects of cutting parameters (MOC-V-f) on tool wear.

The analysis of the effect of each control factor on tool wear was conducted using the signal-to-noise (S/N) ratio response table (Table 7). The highest values (marked with *) in the table indicate the optimal levels. The rank value for the MOC factor is 1, which provides the order of importance of variables. From this table, it can be concluded that the most important factor affecting the results is MOC. This result has been confirmed by the conducted variance analysis. Delta represents the difference between the maximum and minimum values of the respective variable. The largest difference in the delta column indicates that the cooling method is the most important parameter among all controllable parameters. The second most important parameter is feed rate, and the least important parameter is cutting speed.

**Table 7.** S/N ratio response for tool wear (Vb).

| Level | MOC | V | f |
|---|---|---|---|
| 1 | 10.23 | 12.95 * | 14.2 * |
| 2 | 12.79 | 12.88 | 13.15 |
| 3 | 15.06 * | 12.25 | 10.73 |
| Delta | 4.84 | 0.71 | 3.47 |
| Rank | 1 | 3 | 2 |

In Figure 8, the effects of cutting parameter levels on tool wear were determined using the S/N ratio. The optimum levels of cutting parameters are seen to be $A_3B_1C_1$ using the "smaller the better" ratio. The slope of the line clearly demonstrates the influence of each control factor. Considering the highest S/N ratio values, the optimum levels are as follows: Level 3 for MOC (nano-MQL), Level 1 for cutting speed (100 m/min), and Level 1 for feed rate (0.1 mm/rev).

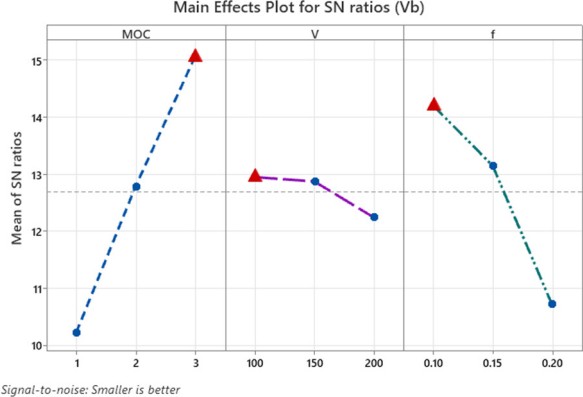

**Figure 8.** S/N ratio for tool wear (Vb).

*3.4. Energy Consumption (Current)*

As a result of the experiments, the lowest current value of 4.18 amperes occurred in Experiment 8, under cooling conditions with nano-MQL, 150 m/min cutting speed, and 0.1 mm/rev feed rate. In the study, the current by the machine's spindle motor was measured. The energy consumption is obtained by multiplying the current, voltage, and cutting time. Therefore, to reduce the processing time, and consequently lower energy consumption, it is necessary to increase the cutting parameters of the cutting and feed rate. Cutting speed and feed rate have a significant impact on power consumption because these parameters reduce processing time. Even though these parameters increase the current value somewhat, they significantly reduce the processing time. Therefore, to achieve minimum power consumption, it is necessary to increase the cutting parameters to reduce processing time [40]. When examining the graphs, it can be seen that the highest current value occurred at the highest values of cutting speed and feed rate under dry-cutting conditions.

According to Figure 9. in dry machining, in addition to its detrimental effect on tool wear, poor surface quality, increased sound intensity, and higher energy consumption in terms of amperage and power were observed in the graphs. MQL and nano-MQL cooling/lubrication applications, on the other hand, are more preferred for surface quality, cutting forces, and tool life [39]. In turning experiments, it can be seen that power consumption decreased with MQL and nano-MQL cooling/lubrication applications compared to dry machining conditions. The lubricant is provided as an aerosol with compressed air or as droplets at the work tool interface. The minimum amount of lubricant used allows the workpiece to be almost dry and chips to form [41]. Regardless of the cooling method, when looking at the graph showing the change in current value with cutting parameters in Figure 9, it can be seen that the steepest slope in the graph occurs at the highest values of

cutting parameters. This indicates that increasing cutting speed and feed rate leads to the highest current value. Despite having the highest electricity consumption per unit time, an increase in these parameters actually means the least consumption since it shortens the total processing time for a certain volume of work.

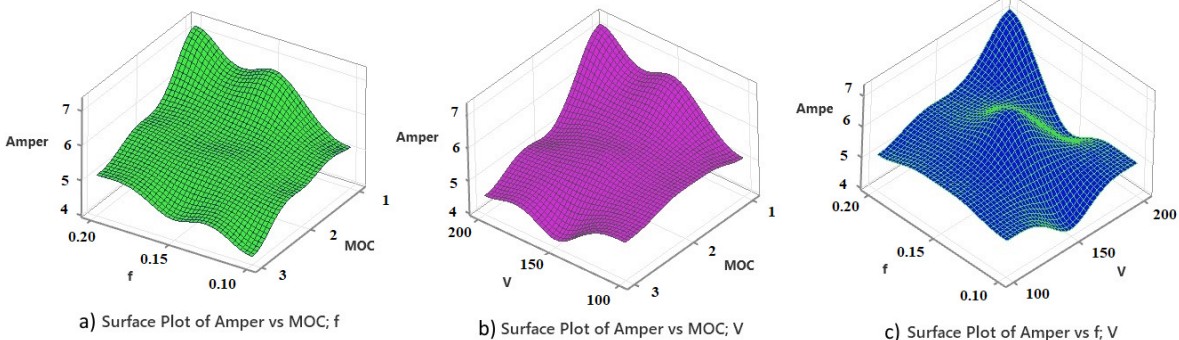

a) Surface Plot of Amper vs MOC; f   b) Surface Plot of Amper vs MOC; V   c) Surface Plot of Amper vs f; V

**Figure 9.** Three-dimensional graphs showing the effects of cutting parameters (MOC-V-f) on current.

The S/N (signal-to-noise) ratios for the changes in current concerning cutting parameters and their levels in the turning process are provided in Table 8. The optimization of measured values and determination of quality characteristics are achieved through S/N ratios. According to the table, the delta value, which has the greatest impact on current variation, is the highest for MOC with 2.71, followed by feed rate with 1.91, and cutting speed with the least effect of 0.39.

**Table 8.** S/N ratio response for current (I).

| Level | MOC | V | f |
|---|---|---|---|
| 1 | −15.94 | −14.37 | −13.54 |
| 2 | −14.58 | −14.64 | −14.76 |
| 3 | −13.23 | −14.75 | −15.45 |
| Delta | 2.71 | 0.39 | 1.91 |
| Rank | 1 | 3 | 2 |

Figure 10 presents the main effect plot of cutting parameters on current. The S/N graph was plotted for the values taken according to the "smaller the better" principle. The highest values in the S/N graph indicate the lowest energy consumption. According to the graph, the optimum points are $A_3B_1C_1$, which means MOC in nano-MQL processing, a cutting speed of 150 m/min, and a feed rate of 0.1 mm/rev. MOC and feed rate have the most significant effect on the current, while the effect of cutting speed appears to be very weak. These results are in line with the studies by Jamil et al. [20].

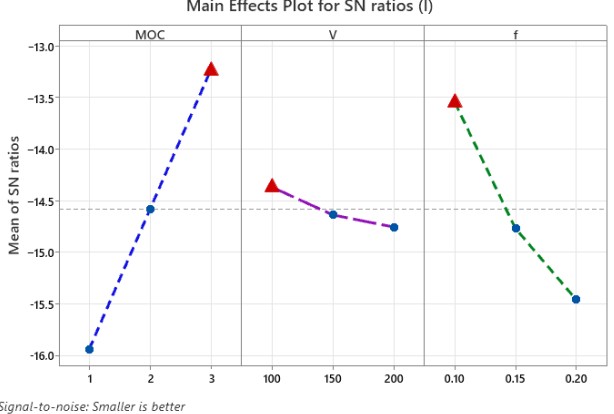

**Figure 10.** S/N ratio for current (I).

### 3.5. Surface Roughness

To determine the surface quality of processed parts, the average surface roughness (Ra) index is generally used. The variation of surface roughness with respect to MOC, cutting speed V (m/min), and feed rate f (mm/rev) is shown in Figure 11. The effect of MOC on other parameters is clearly seen in all three graphs. It is well-known that surface roughness is theoretically a function of feed rate and tool tip radius. Generally, an increase in feed rate leads to an increase in surface roughness [42]. The best surface quality was achieved in Experiment 8, under cooling conditions with nano-MQL, 150 m/min cutting speed, and 0.1 mm/rev feed rate. This situation is particularly evident in the interaction between MOC and V as well as V and f. The lowest Ra values occur in this region of the curve. Similarly, Albayrak et al. have claimed that the most effective parameters for surface roughness are feed rate and spindle speed [26].

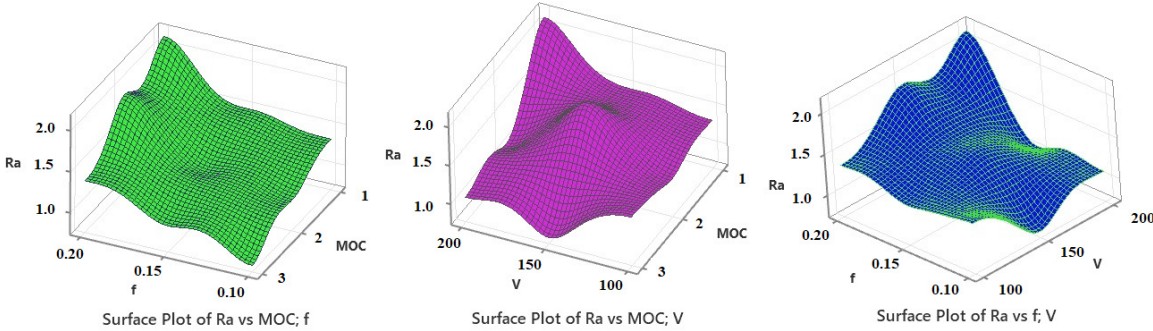

**Figure 11.** Three-dimensional graphs showing the effects of cutting parameters (MOC-V-f) on surface roughness.

In MQL, nano-lubricants create an oxide film in the cutting zone to reduce friction, providing adequate lubrication. Nano-lubricants remove more heat from the heating zone. Therefore, the adhesion between the tool and workpiece material decreases, which helps to keep the cutting edges sharp and reduces cutting forces. The reduction in cutting forces directly implies reduced power consumption [43].

Nanoparticles cause a cushioning effect in the nano-lubricant. The cushioning effect absorbs sudden impacts by reducing fluctuations in cutting force. As a result, a decrease in surface roughness occurs [43].

The results indicate that MQL with nanoparticle-based lubrication reduces roughness values in different combinations of input parameters. Similar results are frequently encountered in the literature.

Dry cutting, MQL, and nano-MQL conditions have shown an increasing trend in surface roughness with increasing cutting speed. This may be related to an increase in cutting temperature with an increase in cutting speed, making it difficult to eliminate or reduce the generated cutting heat, resulting in tool wear and deterioration of the processed surface [11].

The calculated S/N ratios for the control factors used in the turning process regarding surface roughness are provided in Table 9. The analysis of the effect of each control factor on surface roughness was performed using the signal-to-noise ratio response table. Consistent with the literature, the factor with the largest delta value was f = 4.405, making it the most significant parameter. MOC = 3.758 was the second most important parameter, while V = 0.258 was found to be of relatively low importance.

**Table 9.** S/N ratio response for surface roughness (Ra).

| Level | MOC | V | f |
|---|---|---|---|
| 1 | −4.180 | −2.335 | −0.535 |
| 2 | −2.725 | −2.399 | −1.852 |
| 3 | −0.421 | −2.593 | −4.939 |

**Table 9.** *Cont.*

| Level | MOC | V | f |
|---|---|---|---|
| Delta | 3.758 | 0.258 | 4.405 |
| Rank | 2 | 3 | 1 |

In Figure 12, the effects of cutting parameter levels on surface roughness were determined using the S/N ratio. When evaluating the S/N ratios for surface roughness (Ra), it is understood that the optimum cutting condition is $A_3B_1C_1$. Since the slope of the line indicates the power of each control factor's effect, it is clear from the graph that the cutting speed has a very weak effect on roughness, while the most significant effect is with feed rate and MOC as the cooling method. It is also evident from the graph that the highest roughness value is achieved in dry cutting conditions and at the highest feed rate.

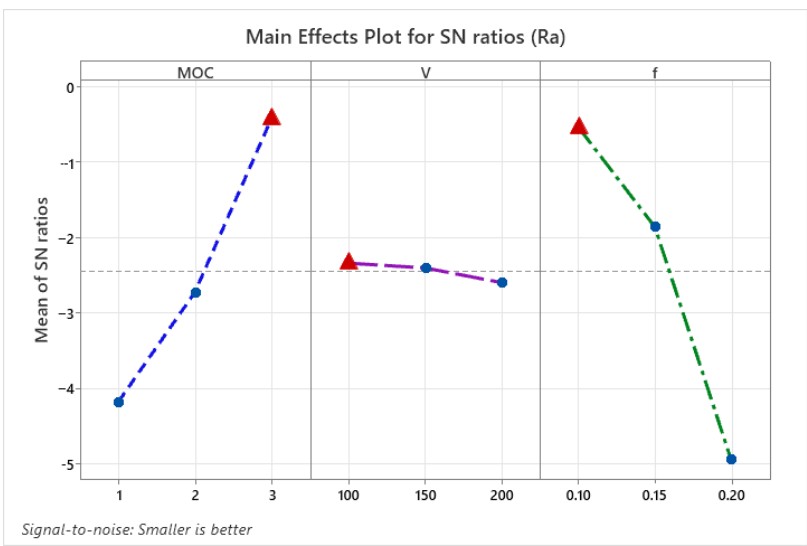

**Figure 12.** S/N ratio for surface roughness (Ra).

*3.6. Sound Intensity*

The level of noise generated during machining with chips is a complex issue influenced by various factors. These factors can arise from many variables such as the machining method used, material type, the condition of the cutting tool, machine type, and operating conditions. In machining with chips, the level of sound increases due to vibrations and noises generated during the cutting and shaping of the material by the cutting tool. The sound level during the process can vary depending on the type and size of the machine used.

Sound intensity was measured instantaneously during the experiments. The measured sound levels result from the interaction between the tool and the workpiece. Sound intensity undergoes an average change of approximately 2–3 dB under the same cutting conditions. Data were obtained at a frequency of 2 Hz with the measuring machine. The sound level associated with that specific experimental condition was determined by averaging all measurements. Three-dimensional graphs drawn based on the data obtained from the experiment results are shown in Figure 13. From all three graphs, it can be observed that the lowest sound intensity occurs under nano-MQL cutting conditions, with the lowest feed rate of 0.1 mm/rev and the middle level cutting speed of 150 m/min. The highest sound level, on the other hand, is observed when the feed rate is at its lowest and under nano-MQL conditions with the highest values for both cutting speed and feed rate. In their study, Albayrak et al. identified spindle speed, feed rate, and chip depth as the most effective parameters for sound level. They concluded that the spindle speed being identified as an important parameter was a result of the machine's inherent sound [26].

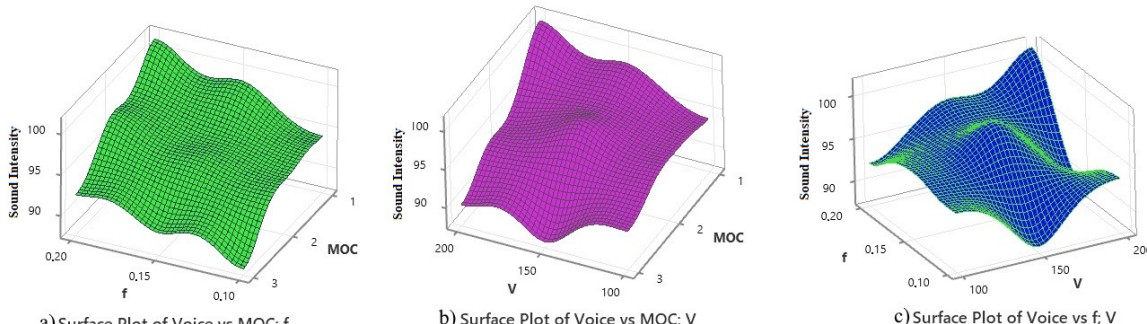

**Figure 13.** Three-dimensional graphs showing the effects of cutting parameters (MOC-V-f) on sound intensity.

From all the graphs, it is evident that the best results are obtained from the cooling model with nano-MQL. Other researchers have also confirmed the dominant role of nano-MQL in sound intensity [44,45].

The findings suggest that the increase in parameters such as tool wear, surface roughness, and current plays a role in increasing cutting sound [46]. This situation has been expressed in other studies as an increase in sound level leading to an increase in machining forces and surface roughness while reducing power consumption [47]. Similarly, an increase in surface roughness of processed parts and an increase in sound pressure levels in the environment have been observed as tool wear increases [48].

The analysis of the interaction of each control factor with sound intensity is shown in Table 10 in the S/N response table. Since the greatest difference between levels indicates the degree of interaction, MOC is the most important factor affecting sound intensity with a difference value of 0.73. Following MOC, feed rate with a difference value of 0.46 and cutting speed with a difference value of 0.09 are of relatively lesser importance. Again, according to the graph, the effects of the factors are very close in value.

**Table 10.** S/N Ratio response for sound intensity (S).

| Level | MOC | V | f |
|-------|-------|-------|-------|
| 1 | −39.83 | −39.45 | −39.27 |
| 2 | −39.6 | −39.52 | −39.52 |
| 3 | −39.1 | −39.55 | −39.74 |
| Delta | 0.73 | 0.09 | 0.46 |
| Rank | 1 | 3 | 2 |

Similarly, the S/N graph showing the interaction between sound intensity and control factors is shown in Figure 14. When evaluating the S/N ratios for sound intensity, it is observed that the optimum cutting conditions are $A_3B_1C_1$. Considering the highest S/N ratios, it can be seen that the optimum levels are the 3rd level for MOC, the 1st level for cutting speed (100 m/min), and the 1st level for feed rate (0.1 mm/rev). It is apparent from the graph that MOC is a dominant factor in sound intensity. As lower sound levels are desired, it is seen that the lowest sound level is achieved under nano-MQL cooling conditions with the lowest cutting speed and feed rate, indicating that cutting speed has a weaker effect compared to other factors.

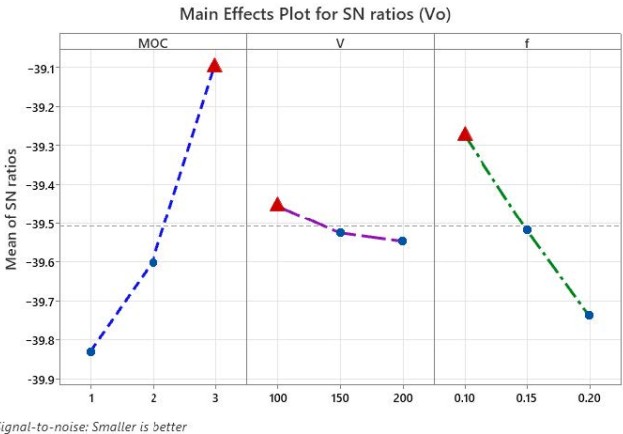

**Figure 14.** S/N ratio for sound intensity (SI).

### 3.7. Investigation of Chip Morphology

During machining with chips, a significant portion of mechanical energy is expended in this region due to the interaction between the workpiece and the tool. The structure of the chips formed during turning can provide important information about cutting processes. Chip morphology, in essence, refers to the physical properties of chips such as size, diameter, shape, and appearance [11]. In turning, chip morphology is primarily influenced by characteristics such as cutting parameters, tool-workpiece interactions, and cooling. These characteristics affect factors such as material quality, cutting speed, tool life, and the surface quality obtained after machining. Therefore, correctly interpreting and understanding chip morphology is essential for a high-quality and efficient turning process.

The chips obtained from the experiments were examined under a microscope to gather information about chip formation mechanisms. In Figure 15, the top part represents the chip thickness obtained from each experiment, while the bottom part provides the overall appearance of the chips. Under dry cutting conditions, continuous chips were obtained at low feed rates, while an increase in the feed rate resulted in more segmented chips. As seen in the figure, under MQL and nano-MQL conditions, a reduction in curling radius and the formation of segmented chips is observed. This might be due to better lubrication in the sliding zone with MQL and the active role of nanoparticles in heat transfer under nano-MQL conditions.

As seen in the chip images, serrated chips were formed under all processing conditions investigated during the turning of AISI 4340. Although the serrated nature of chips varied depending on the cutting conditions, it is believed that this is primarily due to the relatively lower hardness of the material. Palanisamy et al. [16] explained this phenomenon as localized chip formation due to the thermal softening of the material in the cutting zone. The serrations and instability in serrated chips were attributed to local thermal softening in the chips, resulting in significant deformation compared to adjacent materials.

The positioning and angle of the nozzle that delivers the MQL system to the cutting zone also significantly affects chip morphology formation. In the experiments, the nozzle was placed 20 cm away from the working area and directly over the chips, based on the literature. Similarly, in another study conducted, it was stated that this placement of the nozzle not only provided cooling but also lubrication, and it also reduced the formation of chips into small pieces, thus reducing the additional heat generated from friction between the tool and the workpiece [49].

When examining the chips as chip thickness from the images, it is observed that the highest chip thickness occurred in dry cutting, and the lowest chip thickness occurred in nano-MQL cutting. This situation occurs not only with an increase in cutting and feed rates but mainly in dry cutting, due to the inability to remove the heat generated in the cutting zone, leading to thermal softening in both the workpiece and the tool. Chips produced in

dry cutting are wider compared to those produced in MQL cutting. The wider chips in dry cutting are a result of the lateral flow observed in the cutting plane, as previously observed by the authors [50]. When MQL and nano-fluids are introduced, these coolants not only remove heat from the cutting zone but also create a film layer, reducing tool wear and changing the crater angle, leading to the formation of small broken chips. When examining the chip forms in the figure (exp. 1 V = 100 m/min, f = 0.1 mm/rev), it is seen that the curling radii of the chips decreases and chip thickness increases at low cutting speeds and high feed rates [46]. Similar results were found in a study by Khandekar et al. [2].

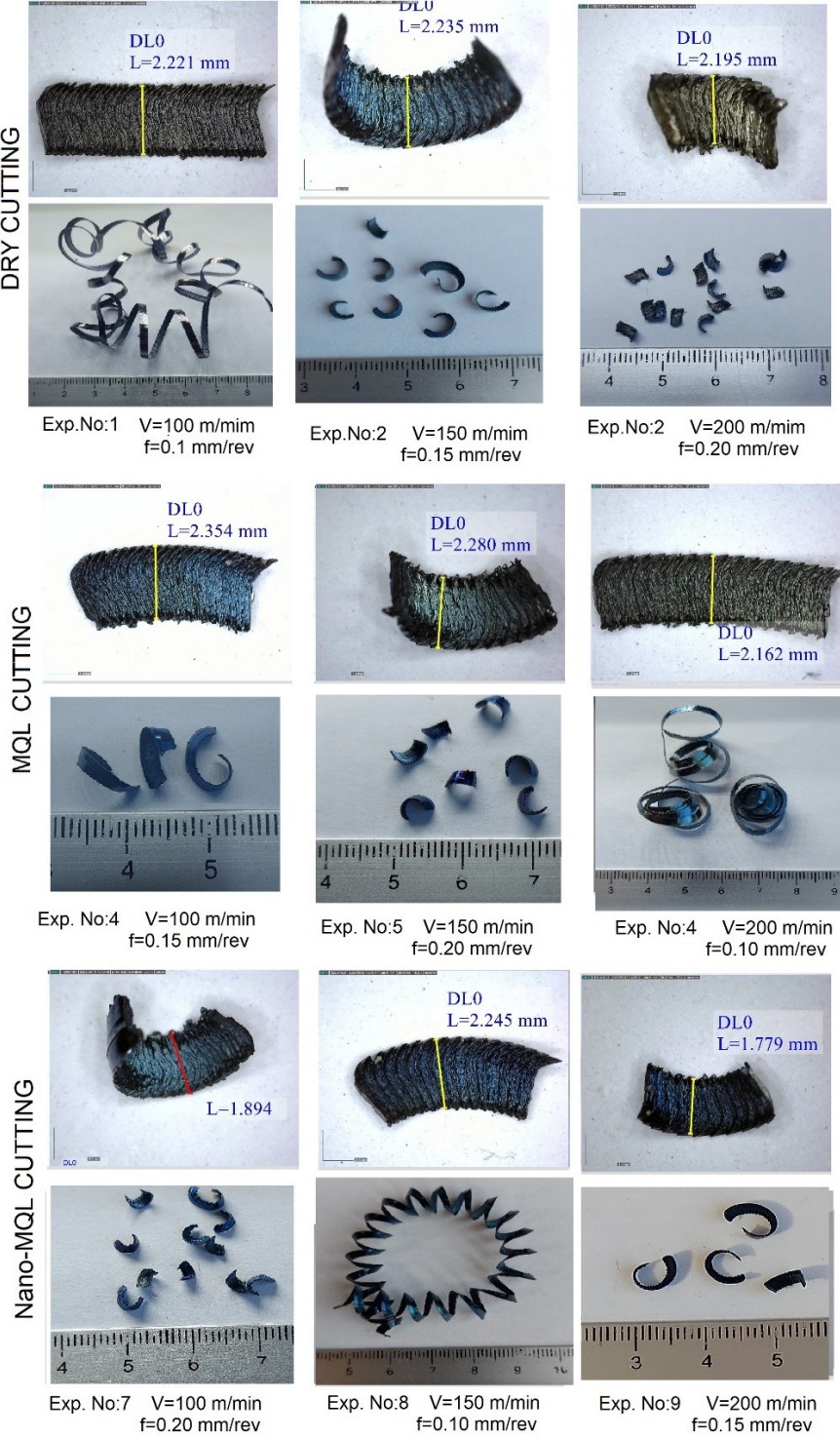

**Figure 15.** Chip morphology under dry, MQL, and nano-MQL cutting conditions.

*3.8. Confirmation Tests*

In the experimental study, optimal results were obtained for tool wear, surface roughness, current, and sound intensity values, and the parameters that had an impact on the results were determined through variance analysis. The final step of the optimization process is to perform confirmation experiments and validate the optimization process. The parameter set that gives optimal values as a result of Taguchi optimization can sometimes be any of the existing experiments, while sometimes it may be an experiment conducted outside of the ones already performed. In this study, both Vb, Ra, I, and SI values reached the optimum result under the $A_3B_1C_1$ experimental conditions, which was different from the existing experiments. Therefore, confirmation experiments were conducted under $A_3B_1C_1$ (nano-MQL, V = 100 m/min, f = 0.1 mm/rev) conditions determined through Taguchi S/N optimization. The results obtained are compared with the results obtained from the experiments in Table 11. This parameter set was not part of the conducted L9 experimental set. Therefore, control experiments were carried out under the conditions determined by Taguchi S/N optimization, namely $A_3B_1C_1$ (nano-MQL, V = 100 m/min, f = 0.1 mm/rev). The obtained results were compared with the results from the experiments in Table 11. The verification test was performed three times, and the average of the data was taken. As a result of the verification test, Vb = 0.13 mm, I = 3.85 amperes, Ra = 0.73 μm, and SI = 81.3 dB values were obtained. When comparing these results with the results obtained according to Taguchi L9 S/N, it was observed that the values were 13% better for Vb, 7% better for I, 9% better for Ra, and 8% better for SI.

**Table 11.** Confirmation test results.

|  | **Vb** | **I** | **Ra** | **SI** |
|---|---|---|---|---|
| Exp. 8 ($A_3B_2C_1$) | 0.15 | 4.18 | 0.81 | 88.1 |
| Exp. ($A_3B_1C_1$) | 0.13 | 3.85 | 0.73 | 81.3 |
| Percentage Improvement | 13 | 7 | 9 | 8 |

## 4. Conclusions

In this study, AISI 4340 alloy steel was subjected to turning under different cooling conditions using PVD-coated tools. Cutting parameters' effects on tool wear, surface roughness, the current by the CNC, and ambient sound intensity were investigated in dry, MQL, and nano-MQL cutting processes.

The results obtained from the experiments can be summarized as follows:

(1) The best values for all output parameters were achieved in experiment number 8 under the $A_3B_2C_1$ conditions, with Vb = 0.15 mm, Ra = 0.81 μm, sound intensity of 88.1 dB, and a current value of I = 4.18 A;

(2) The effects of cutting parameters on response variables were analyzed using the statistical method of ANOVA. The percentage contributions to the variation were as follows: MOC was the most influential factor with 55.97% for tool wear, 59.94% for current, 54.89% for surface roughness, and 67.27% for sound intensity. In this study, tool wear was modeled with an accuracy of 92.10%, current at 92.00%, surface roughness at 92.50%, and sound intensity at 95.87%;

(3) Flank wear was measured as tool wear using a microscope. Burn marks were generally observed on the cutting tools due to excessive heating. There was generally no sign of chipping or fracturing on the cutting edges, and uniform wear was observed in all tools. The highest wear occurred in experiment number 3, with Vb = 0.420 mm, under dry cutting conditions with V = 200 m/min and f = 0.2 mm/rev, while the lowest wear was observed in experiment number 8, with Vb = 0.150 mm, under nano-MQL conditions with V = 150 m/min and f = 0.1 mm/rev.

(4) Cutting parameters' effects on tool wear, current, surface roughness, and sound intensity were determined using the Taguchi S/N ratio. According to the S/N ratio, the optimum cutting conditions were found to be $A_3B_1C_1$ for all output parameters. Since these experimental conditions were not part of the L9 series, confirmation experiments

were conducted under nano-MQL cooling and V = 100 m/min, f = 0.1 mm/rev cutting conditions. The results of the confirmation experiment showed an improvement of 13% in tool wear, 7% in current, 9% in surface roughness, and 8% in sound intensity compared to the normal experimental results.

(5) When examining chip morphology, continuous chips with low feed rates and cutting conditions were obtained under dry cutting conditions, while with MQL and nano-MQL cooling, segmented chips with smaller radii were obtained at higher speeds;

(6) The use of nano-MQL coolant resulted in the lowest values for all output parameters. In dry cutting conditions with high cutting speeds and feed rates, tool wear increased due to excessive heating in the cutting zone. An increase in tool wear led to higher values for sound intensity and surface roughness. As the required cutting force increased, the current also tended to increase. There was a positive relationship between surface roughness, tool wear, sound intensity, and current. When one of these four values increased, the others also increased.

In conclusion, nano-MQL systems appear to have several advantages among the cooling/lubrication methods used in machining. However, during the experiments, it was observed that MWCNT, used as nanomaterial, had a tendency to adhere to all environments and atomize in the MQL spraying system, especially affecting the operator's health. For the sustainability of the necessary environmental impact conditions, future studies should pay attention to these issues. Along with the resolution of such problems, it is anticipated that the nano-MQL system will contribute to environmental friendliness, cleaner production, and the improvement of desired machinability properties.

There are many studies in the literature on the optimization of tool wear and surface roughness, which form the basis of the cutting parameters of AISI 4340 steel. However, in addition to these, in terms of sustainability, energy efficiency under different cooling/lubrication conditions and reduction in sound emission (sound pollution) are important parameters in terms of green production. This study aims to analyze the effects of dry, MQL and nano-MQL conditions on the output parameters of AISI 4340 steel, which has an intensive use in dry, MQL and nano-MQL conditions, and to suggest the optimum parameters for researchers and the use of steel in industry. Again, different chip morphology outputs will also be helpful for those who will work on this subject.

**Funding:** This research received no external funding.

**Data Availability Statement:** Not applicable.

**Conflicts of Interest:** The authors declare no conflict of interest.

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
