# Peer review of "Exploring the Impact of the Turning of AISI 4340 Steel on Tool Wear, Surface Roughness, Sound Intensity, and Power Consumption under Dry, MQL, and Nano-MQL Conditions"

_lubricants, doi:10.3390/lubricants11100442_

Round 1
Reviewer 1 Report
1. The author states that optimizing input parameters (in this case cooling condition, speed and feed) will help improve production efficiency, which is correct. However, the abstract gives an impression as though, the main objective is to optimize the output parameters. Especially, the statement - “the optimal output values are found as Vb=0.15 mm, Ra=0.81μm, 16 88.1 dB for sound intensity, and I=4.18 A for current”.
2. Page 2, line 25 - what does the author mean by “chip manufacturing”? It would be better to replace this with “machining processes” or some other suitable word.
3. Again at the end of the introduction section, the author states “to optimize tool wear, power consumption, sound levels reduction, and surface roughness”. The objective is to optimize the input parameters? All the output parameters have to be reduced / lowered to improve machinability. Is it not?
4. Was any surfactant added to the coolant to reduce agglomeration of the nano-particles?
5. In line 306-307, it is wrongly stated “maximize the surface roughness Ra for processing efficiency”. The objective is to minimize surface roughness.
6. Units for Vb, Ra, SI and I to be provided in table 5. Also, Max is mis-typed as “Mak” in the last row of the table 5.
7. In the ANOVA table 6, R2 is typed without superscript as R2. Also maintain consistency within the table. In two places, R-sq is used. In two places R2 is used.
8. How is this statement given from figure 7? “the feed rate has the highest effect on tool wear, especially under dry cutting conditions”. Along with feed, cutting speed also appears to have high effect on tool wear from figure 7. Please comment. The same linear relationship is seen for both feed and cutting speed because in the Taguchi experimental combination table 4, it can be seen that for dry experiments - 1,2 and 3, both feed and cutting speed were varied together.
9. Will the clamp meter provide continuous measurement of current values? How will the current values which vary during the turning process be recorded? Is the current value which is reported in the contour plot in fig. 9 obtained from a single measurement when the clamp meter is clamped onto the power cable?
Author Response
The answers given to the reviewer are in the attached file.

Reviewer 2 Report
Researcher adopted the control parameters in the experiments are three different cooling types (dry, MQL and MWCNT additive nano-MQL, cutting speed, and feed rate. The responses are tool wear, power consumption, sound levels reduction, and surface roughness. The article is very interesting.
- How author selected the the workpiece dimension as 150 mm in length and Ø260 mm in diameter.
- Provide the reference for The chemical properties of the AISI 185 4340 alloy steel used in the experiments are provided in Table 1.
- Provide the reference for the mechanical properties are given in Table 2
- Provide the full form of Vb, Ra and I at its first appearnace in the abstract section.
- Authors mentioned that "AISI 4340 alloy steel, environmental consciousness in line with green manufacturing, the impact of sound during machining on both human health and cutting parameters, wear, surface roughness, and energy consumption" ? Justify ? What is green manufacturing?
- Provide the scientific reason why reseachers used PVD-coated cutting inserts of TaeguTec brand, DNMG 150608 198 TT5080 type? Why not CVD coated or Uncoated Tool.
- In accordance with the literature, a flow rate of 100 mL/h was used in the experimental study. Provide literature support in this context.
- Check the Figure 4. Experimental setup of the study? An unidentified symbol has been shown. Try to remove it. (CNC Lathe machining). Here change the word voice with sound intensity.
- Authors adopted Taguchi L9(33 ) ortogonal array. Provide reference.
-Provide the reason for selecting higher cutting speed as 200 m/min.
- Table 6. Variance Analysis (ANOVA) for Turning Responses. Use a column pointing towards significant/ insignificant term.
- What is R2? R-Squared (R² or the coefficient of determination). Write properly.
- What is the signifiance of S in Table 6.
- What is MOC? Provide the full form in its first appearance.
- . These values being greater than 85% indicate that the regression model fits well, and models close to reality can be obtained. Provide citation. (Line no. 397-399)
- Explain Table 14. Confirmation test results
- As a result of optimization, 21 it was concluded that the optimum values for cutting conditions (A3B1C1) were nano-MQL cooling 22 and V=100 m/min, f=0.1 mm/rev cutting. How optimum values has been obtained?
-Why author adopted First-degree regression?
- What is the novelty of this work?
- Refer the following papers
Molnar, V. (2023). Experimental Investigation of Tribology-Related Topography Parameters of Hard-Turned and Ground 16MnCr5 Surfaces. Lubricants, 11(6), 263.
Mallick, R., Kumar, R., Panda, A., & Sahoo, A. K. (2023). Hard turning performance investigation of AISI D2 steel under a dual nozzle MQL environment. Lubricants, 11(1), 16.
-
Author Response

(The authors gave the same response as above.)

Reviewer 3 Report
In this work, the different lubrication conditions and control parameters were studied to screen the optimal process to improve the machining properties of AISI 4340 steel. The research content of this paper is substantial, the research is suitable for practical application, and has high engineering reference value. A lot of statistical analysis and comparative analysis are carried out in this paper, but the scientific mechanism is not discussed enough. Therefore, the manuscript needs further polishing and necessary explanations need to be added in order to meet the bar for publication. Specific comments are included below.
Major:
1. It is best not to use abbreviations directly in the abstract, such as (MQL), (A3B1C1), etc. After all abbreviations are first proposed in the full text, please maintain the unity used in the following paragraphs, do not repeatedly use the full name and abbreviation, please revise.
2. The manuscript is too verbose and ill-structured. Especially in the “Introduction”, a large number of references and repeated narration of the research background of different research projects have led to the serious weakening of the research highlights, innovation points and significance of this paper. For example, in the second paragraph of the “Introduction”, there are contradictions in the description. All this makes the readers think that this study only obtained a set of optimal process parameters through experimental testing. Please delete some unnecessary information in the “Introduction”, simplify the conclusions of relevant research, and highly condense the research focus and significance of this paper.
3. The same problems appeared in the “Result and Discussion” section. The author has carefully explored the influence of multiple influencing factors on the machinability of AISI 4340 steel, which is worthy of approval. It is suggested that the author merges together to discuss the influence of different parameters on “Tool Wear”, “Energy Consumption”, “Surface Roughness” and “Sound Intensity” together to reduce the repetition. Or repolish the language.
4. If possible, use a, b, c to label the Figures.
Minor:
5. 3.2. Analysis of Variance: It is suggested that the data obtained from statistical analysis be made into bar charts or line charts to make the results more intuitive.
6. Lines 418-428: “In dry cutting conditions, as the feed rate 422 increases, wear exhibits a linear increase”, The linear relationship is wrong, the author does not give evidence of linear relationship. The two are positively correlated.
7. Line 445 and 491: Please explain "the smaller the better".
8. Figure 7, 9 and 11: 3D Graphs all showed that the cutting parameter results were optimal when V=150 m/min. Why is it inconsistent with the research conclusion of the manuscript? Please explain the basis for judgment.
The language needs to be further refined and the level of expression needs to be further clarified.
Author Response

(The authors gave the same response as above.)

Round 2
Reviewer 1 Report
The author has provided the necessary remarks for the queries raised during the previous review. The corrections carried out have also been done to meet the necessary standard of the Journal.
Reviewer 3 Report
Accept the author's response.
Some figures are still not marked with figure numbers (a, b, c), please consider marking.